# Applications of Smartphone-Based Aptasensor for Diverse Targets Detection

**DOI:** 10.3390/bios12070477

**Published:** 2022-06-30

**Authors:** Ying Lan, Baixun He, Cherie S. Tan, Dong Ming

**Affiliations:** 1Academy of Medical Engineering and Translational Medicine, Tianjin University, Tianjin 300072, China; ly13207642019@tju.edu.cn (Y.L.); baixun_he@tju.edu.cn (B.H.); 2Tianjin Key Laboratory of Brain Science and Neuroengineering, Tianjin 300072, China; 3Department of Biomedical Engineering, College of Precision Instruments and Optoelectronics Engineering, Tianjin University, Tianjin 300072, China

**Keywords:** aptamer, aptasensor, biosensor, smartphone, POCT

## Abstract

Aptamers are a particular class of functional recognition ligands with high specificity and affinity to their targets. As the candidate recognition layer of biosensors, aptamers can be used to sense biomolecules. Aptasensors, aptamer-based biosensors, have been demonstrated to be specific, sensitive, and cost-effective. Furthermore, smartphone-based devices have shown their advantages in binding to aptasensors for point-of-care testing (POCT), which offers an immediate or spontaneous responding time for biological testing. This review describes smartphone-based aptasensors to detect various targets such as metal ions, nucleic acids, proteins, and cells. Additionally, the focus is also on aptasensors-related technologies and configurations.

## 1. Introduction

Biosensors can recognize the target analyte or a specific group of target molecules in complex samples [1]. A biosensor comprises three parts: molecular recognition elements (such as nucleic acids, receptors, antibodies, enzymes, organelles, tissues, microorganisms, and animal or plant cells, among others) [2], a transducer, and a signal processing unit. With the advancement of science and technology, recently developed biosensors are more portable, rapid, and user-friendly [3]. Hence, biosensors have been widely applied in the food industry [4,5], environmental monitoring [6,7], clinical monitoring [8,9], biomedicine [10,11], and other fields.

Given the rapid advances in personalized medicine and portable health care, point-of-care testing (POCT) based on biosensors has excellent potential. Advances in nanomaterials, molecular biotechnology, and optoelectronic integration have promoted the development of biosensing toward low-cost, easy-to-use, and on-site POCT [12]. Earlier POCT usually required high-cost and complex peripheral devices for analysis and evaluation, limiting its global and public health applications [13]. Therefore, POCT with low cost and easy operation urgently needs to be developed. Smartphones can solve these problems effectively solved by the integration of sensing, processing, and connectivity. Smartphones appeared in the consumer market in the late 1990s and quickly replaced the mobile phone market [14]. Typically, combining basic features such as voice and messaging with powerful computing technology, smartphones can support third-party applications, sensing, Internet access, and wireless connections with other devices [15]. Furthermore, smartphones can also equip with various sensors for various applications in multiple fields and providing substantial resources [16]. Additionally, smartphones with powerful computing and analysis functions can take high-resolution images, which have the characteristics of image storage, portability, and easy accessibility [17]. Hence, many scientific research institutions integrate smartphones into detection systems, such as colorimetric [18], electrochemical [19,20], optical [21,22], and other aspects of signal processing and image acquisition.

Aptamers are short oligonucleotide sequences obtained by in vitro screening, which can bind to multiple targets with high affinity and specificity, and have unique physical and chemical characteristics [23]. Aptamers with stable performance can specifically bind small substances such as ions [24,25], molecules [26], proteins [27,28], and cells [29]. Aptamers can be selected by the systematic evolution of ligands by exponential enrichment (SELEX), which is an iterative process that involves repetitive cycles of binding, partitioning, and amplification steps. The new SELEX strategies were developed to simplify procedures and improve aptamers species [30], such as microfluidic SELESX, in silico SELEX, bead-based SELEX, in vivo SELEX, and so on. As the most effective bioreceptor sensing component, the aptasensors become a powerful sensing instrument [31]. Aptamer has been an attractive tool for various applications compared with antibodies and enzymes due to its cost-effectiveness, strong specificity, suitable pretreatment, simple operation, and short detection time [32]. Recently, aptasensors have been applied to cell imaging [33,34], drug delivery [35,36], disease treatment [37,38], microbial detection [39,40], and other fields. Hence, a smartphone-based aptasensor will be an ideal tool for POCT and analysis to obtain a fast and precise response in various fields. This review summarizes the advancements in smartphone-based aptasensor, emphasizing various detection methods of aptasensors for the different target analytes.

## 2. Smartphone-Based Aptasensor

Smartphone-based aptasensors have been demonstrated to detect five types of targets: metal ions, small molecules, nucleic acids, proteins or glycoproteins, and bacteria. Furthermore, multiple-type targets can also be detected by a multi-channel system simultaneously.

### 2.1. Metal Ions Detection

Heavy metals are non-biodegradable and ubiquitously distributed in the biosphere with a highly toxic property [41]. Heavy metal ions discharged into the environment from household and industrial wastewater are likely to accumulate in living organisms through the alimentary canal, thereby harming human health [42,43]. Therefore, developing a fast and sensitive heavy metal ions detection method is very urgent and essential.

Many studies have reported the detection of heavy metal ions such as Cd^2+^ and Hg^2+^ based on their specific aptamers [44,45,46,47]. For Cd^2+^ detection, Wu et al. [44] presented a SELEX strategy for selecting Cd^2+^ aptamers and further adopted the selected aptamer as a recognition molecule to obtain the colorimetric detection of Cd^2+^. To improve the convenience and rapidity of detection, Gan et al. [48] introduced a smartphone-based colorimetric system to detect Cd^2+^. As shown in Figure 1a, in the presence of Cd^2+^, Cd-aptamers tend to combine with Cd^2+^ rather than gold nanoparticles(AuNPs), causing the dispersion of AuNPs. After adding NaCl solution, the AuNPs could aggregate to a different degree due to the number difference of bare AuNPs, making the solution changeable in color. While, in the absence of Cd^2+^ in the sample, aptamers immobilized on the AuNPs protected AuNPs from aggregation under NaCl conditions. Hence, the color of the last reaction solution changed from red to purple and finally to gray as the Cd^2+^ concentration increased. As shown in Figure 1b, a self-developed smartphone-based colorimetric system was used to capture the colorimetric image of a 96-well microplate, extract the pixel of the image, and then analyze the image data by the image processing algorithm. At last, the concentrations of Cd^2+^ in the sample were calculated by comparison with the colorimetric response standard curve of the Cd^2+^-spiked tap water sample. The linear range of Cd^2+^ was 2–20 μg/L, while the limit of detection (LOD) was 1.12 ug/L, showing a high selectivity and sensitivity of the smartphone-based colorimetric system.

Employing the same colorimetry principle based on the color change of AuNPs after adding NaCl solution, Xiao et al. [49] and Sajed et al. [50] proposed distinct smartphone-based aptasensors to detect Hg^2+^ or Cd^2+^. As depicted in Figure 1c, Xiao et al. [49] designed a microwell reader consisting of a resistor, a 520 nm light-emitting diode (LED), and a microwell to detect Hg^2+^. The microwell reader containing the sample was positioned on the top of the ambient light sensor. Following pressing the switch to form exciting light, the transmitted light intensity data were recorded and displayed on the light meter app, and the concentrations of Hg^2+^ were determined with the linear range of 1–32 ng/mL, and the LOD of 0.28 ng/mL in both Pearl River water and tap water samples.

Subsequently, Sajed [50] et al. used colorimetry at multiple points of the visible spectrum (470, 540, 640 nm). The group fabricated a gadget using three-dimensional printing technology applicable to any type of smartphone and integrated it with optical components, as revealed in Figure 1d. The gadget consisted of a thin film transistor liquid crystal display (TFTLED) board chamber, a focusing chamber of the camera, and two cuvette chambers containing the reference solution and the sample solution. A full-color TFTLCD was adopted as the light source, providing a uniform irradiance condition with any predefined pattern. Hg^2+^ concentration was estimated using multiple linear regression (MLR), a statistical technique for analyzing the connection between a dependent variable and multiple independent variables. In the MLR model, red, green, and blue (RGB) values extracted from images of samples by smartphone were applied as the input, while the concentration of Hg^2+^ in the sample was applied as the regression target. Light source enhancement, RGB analysis, and a novel image processing protocol make it possible to achieve an excellent level of sensitivity (1 nM).

Apart from NaCl solution, polydiene dimethyl ammonium chloride (PDDA) also leads to the aggregation of AuNPs, which can make the color of the response solution shift reported by Xu et al. [51]. The aptasensor combined with smartphone colorimetry to detect Cd^2+^ based on an aptamer competitive binding assay of PDDA and Cd^2+^. The directional light on the sealed box is reflected on the measured object by aluminum foil, as shown in Figure 1e. The images of samples were captured by smartphone, and a ColorAssist app read the RGB values. The linear correlation between Cd^2+^ concentration and red value (R-value) was plotted in Figure 1e, with a linear range of 1–400 ng/mL. The aptasensor of this work showed the potential for practical application as its results were in accordance with those from the atomic absorption spectrometer.

### 2.2. Small Molecules Detection

Many studies have reported small-molecule assays based on specific aptamers, such as cyanotoxin [52,53,54], kanamycin [55,56,57], and ATP [58]. In recent years, with an emphasis on POCT detection, many researchers have combined aptamer and smartphones to identify small molecules. Table 1 summarizes small-molecule detection using smartphone-based aptasensors. For example, Li et al. [59] developed an easy-to-use and miniaturized detector, using a fluorescent aptasensor array composed of specific aptamers and dyes for the multiplexed detection of four common cyanotoxins as environmentally hazardous substances. As shown in Figure 2a, a blue 462 nm laser illuminates single-stranded DNA (ssDNA) dyes without cyanotoxins, and the green fluorescence decreases upon adding analyte solutions, resulting from aptamer conformational changes. The authors also developed a smartphone-based fluorescence readout platform that integrated a microfluidic chip where the aptamer binding assays occurred with a 3D-printed attachment (Figure 2b). The emitted green fluorescence from the microfluidic chip was filtrated by a filter in the attachment and detected by the smartphone camera. In the raw images, the fluorescent intensity of the reaction was reversely proportional to the concentration of anatoxin-a (ATX). At the same time, the other four areas did not show any distinct changes in fluorescence signals, which certified the high specificity of aptamers (Figure 2c). As a result, the smartphone-based sensor platform realized accurate detection and measurement of four common cyanotoxin species, with the LOD lower than 3 nM, in parallel.

In addition to the change in intensity of single-color fluorescence, the analytes can be detected by the signal conversion of two-color fluorescence. Saurabh et al. [60] demonstrated ratiometric Förster resonance energy transfer (FRET) between fluorescent dye pairs on aptamers for rapid-response and sensitive kanamycin detection. As depicted in Figure 2d, the upper half of dye-labeled kanamycin DNA aptamer (KBA) is the kanamycin binding aptamer, while the bottom is double-stranded DNA used for surface modification of glass slide. When kanamycin was bound to KBA_FRET_, KBA displayed a lower level of FRET efficiency due to the unique placement of a donor-acceptor dye pair. The ratiometric FRET-based estimation could be implemented on a custom-made smartphone-based fluorescence setup with a LOD of 28 nM (Figure 2e).

Another energy transfer model superior to FRET is nanometal surface energy transfer (NSET), which has a higher energy transfer efficiency and a more extended energy transfer distance than dipole-dipole FRET [61,62]. Employing the NSET strategy, Nie et al. [63] developed an electrochemiluminescence (ECL) biosensor for point-of-care detection of ATP. As shown in Figure 2f, Fe_3_O_4_ NP@ZIF-8 was used as an effective catalyst, and its poor electron transfer capabilities could not influence the other reagents. MoS_2_ QDs produced NSET with AuNPs by the specific binding of cDNA and ATP aptamer, which decreased the ECL signal of MoS_2_ QDs. When ATP bound to the aptamer, the ECL-NSET system was deconstructed, resulting in the recovery of the ECL signal. Furthermore, a self-developed software for smartphones could capture changes in the ECL signal, which achieved point-of-care detection of ATP with a LOD of 0.015 nmol L^−1^ (Figure 2g).

**Table 1 biosensors-12-00477-t001:** A summary of small-molecule detection using smartphone-based aptasensors.

Target Analyte	Detection Probe	Detection Method	LOD	Reference
Cyanotoxins	aptamer−dye	Fluorescence detection	<3 nM	[59]
Kanamycin	aptamer-dyes	Fluorescence detection	28 nM	[60]
ATP	aptamer-MoS_2_ QD	Electrochemiluminescence	0.015 nmol L^−1^	[63]
ATP	aptamer-Fe(CN)_6_^3−^	Colorimetry	NR	[64]
Mycotoxin	aptamer−dyes	Fluorescence detection	0.1 ng/mL	[65]
Chloramphenicol	AuNPs-aptamer	Colorimetry	5.88 nM	[66]
17-β-estradiol	dye-split aptamer fluorescent beads	Fluorescence detection	1 pg/mL (in spiked wastewater samples)	[67]
Streptomycin	aptamer−dye	Fluorescence analysis	94 nM	[68]
Streptomycins	AuNPs-aptamer	Colorimetry	12.3 nM	[69]
Ibuprofen	AuNPs-aptamer	Colorimetry	1.24 pg/mL (S-Ibu) 3.91 pg/mL (R-Ibu)	[70]
Sulfadimethoxine	AuNPs-aptamer	Colorimetry	0.023 ppm	[71]
Cocaine	AuNPs-aptamer-UCNPs	Luminescence detection	10 nM (in aqueous solution) 50 nM (in human saliva)	[72]

**Figure 2 biosensors-12-00477-f002:**
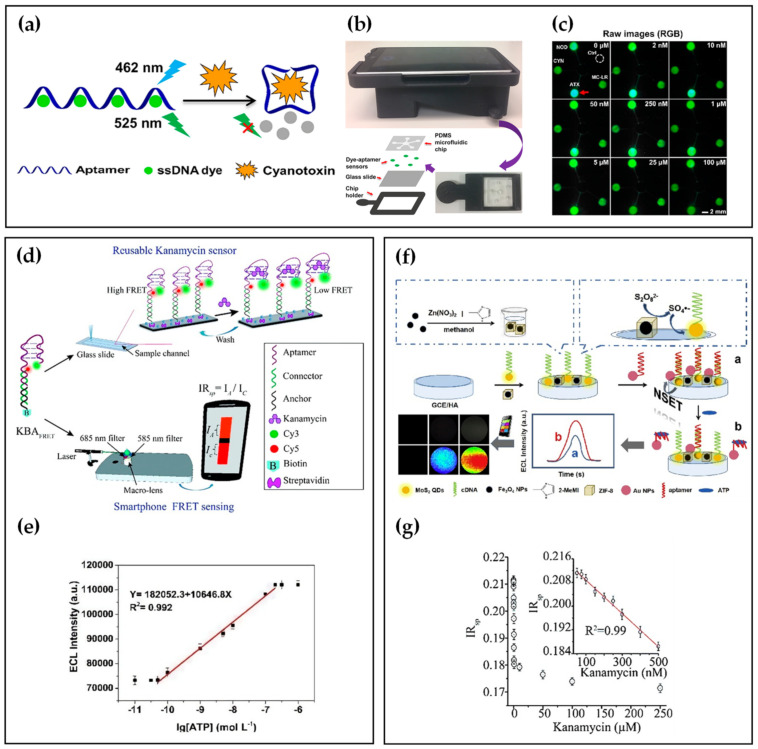
(**a**) Detection theory of the fluorescent aptasensor. Reprinted with permission from Ref. [59], copyright 2019, American Chemical Society. (**b**) Schematic illustration of the smartphone−based fluorescent aptasensor. Reprinted with permission from Ref. [59], copyright 2019, American Chemical Society. (**c**) Raw fluorescence images in response to different concentrations of ATX. Reprinted with permission from Ref. [59], copyright 2019, American Chemical Society. (**d**) Schematic of FRET-based kanamycin aptasensor. Reprinted with permission from Ref. [60], copyright 2019, Royal Society of Chemistry. (**e**) Calibration curve for Kanamycin detection. Reprinted with permission from Ref. [60], copyright 2019, Royal Society of Chemistry. (**f**) Schematic of the ECL biosensor. Reprinted with permission from Ref. [63], copyright 2020, Elsevier. (**g**) Correlation between the ATP concentration and ECL intensity. Reprinted with permission from Ref. [63], copyright 2020, Elsevier.

### 2.3. Nucleic Acids Detection

The principle of detecting nucleic acids using aptamers is complementary base pair, and then specific aptamers can be designed by target sequences [20,73]. MicroRNAs (miRNAs), a kind of nucleic acid, can play as biomarkers of some tumors and cancers [74]. Hence, realizing portable and efficient on-site miRNAs detection is critical for clinical prevention and therapy of many cancer patients [75,76]. Aptasensors for miRNA detection offer higher sensitivity, lower costs, and require less sample than other approaches, opening new possibilities for detecting circulating miRNA in POCT [77,78]. Furthermore, smartphones with portability, high-resolution camera, a global positioning system (GPS), and Internet connection capability can provide healthcare diagnostics in resource-limited circumstances.

Consequently, the combination of smartphone and molecular diagnostics is particularly critical. Although the achievements in this research direction are relatively few, some progress has been made. For example, in the study presented by Shin Low et al. [20], aptamer appeared as a complementary strand of miR-21 binding with each other by complementary base pairs. Then, circulating miR-21 was detected in saliva using a smartphone-based biosensing system. The system consisted of a disposable electrochemical aptasensor, a self-designed circuit board, a smartphone with Bluetooth function, and a specially developed Android application (Figure 3a). Furthermore, the biosensing system demonstrated equivalent performance to a commercial electrochemical workstation, suitable selectivity, and an acceptable recovery rate of 96.2% to 107.2% in spiked artificial saliva.

Tian et al. [79] presented a smartphone-based detection platform for the detection of multiple miRNAs simultaneously. The quantitative detection platform included number-coded hydrogel microparticles, a microscope, and a smartphone with fluorescence image recognition software (Figure 3b). After the reaction shown in Figure 3c, hydrogel microparticles prepared by flow lithography technology displayed different fluorescence intensities. Images were achieved by a smartphone camera and then analyzed by an image processing procedure installed on the smartphone. In this work, the LOD of the detection platform reached the femtomole level in spiked healthy human serum samples, which demonstrated the POCT of tumor biomarkers.

In addition to playing as biomarkers, nucleic acids can constitute viruses. Type A influenza viruses are the most notorious and virulent human pathogens with high morbidity and mortality, which can cause an epidemic or even pandemic [80,81,82]. Lai et al. [83] developed a microfluidic system to screen a specific H1N1 virus-aptamer automatically and efficiently based on the SELEX technique. Furthermore, the selected aptamer completed specific and sensitive detection of the influenza A (H1N1) virus in biological samples by a magnetic-bead assay. Subsequently, Lee et al. [73], based on a specific aptamer, employed a built-in camera of a smartphone and a low-cost homemade setup (Figure 3d) to identify the H1N1 virus successfully with a LOD of 70 ng mL^−1^ in spiked phosphate-buffered saline (PBS) solution. Compared to sophisticated and bulky instruments, this portable device allows users to measure targets anytime and anywhere.

### 2.4. Proteins or Glycoproteins Detection

Carcinoembryonic antigen (CEA), a 180 kDa glycoprotein and a tumor biomarker, plays a significant role in predicting, prognosis, and tumor screening of many malignancies [84,85]. The CEA concentration of healthy individuals is less than 5 μg L^−1^, while patients containing cancer cells are more than 20 μg L^−1^ in serum [86]. Therefore, a rapid, reliable, and sensitive CEA detection method is of particular need to the public. Shu et al. [87] established an electrochemical aptasensor to detect CEA. Based on the signal amplification of AuNPs, the biosensor demonstrated excellent sensitivity and selectivity toward CEA, which indicated a high potential for practical application. Several years later, combining with a smartphone, Qiu et al. [88] developed a paper-based analytical device (PAD) for the fluorescence detection of CEA. As depicted in Figure 4a, the assay was conducted in a centrifuge tube where glucose molecules were gated into the mesoporous silica nanocontainers (MSNs) with the help of the CEA aptamer. Upon adding the analyte, CEA reacted with aptamers contributing to opening the pore and releasing glucose. The released glucose was catalyzed by glucose oxidase (GOD) immobilized on paper to produce hydrogen peroxide and gluconic acid, which quenched the fluorescence of CdTe/CdSe QDs and then measured by a smartphone and a commercial fluorospectrometer. Under optimal conditions, the system enabled sensitive detection of target CEA with a LOD of 6.7 pg mL^−1^ in spiked PBS solution, lower than most commercial enzyme-linked immunosorbent assay (ELISA) kits for CEA.

Enzymes, most of which are proteins, can be detected using aptasensor such as Plasmodium falciparum glutamate dehydrogenase (PfGDH) [89,90] and creatine kinase isoenzymes (CK-MB) [91]. To further increase the convenience of detection, Sanjay et al. [92] constructed a multi-channel, smartphone-based optic fiber platform for quantitative detection of PfGDH. As depicted in Figure 4b, the smartphone flashlight and camera were coupled to two ends of optic fiber by a four-channel smartphone accessory. U-bent optic fiber probes were prepared by gold-plating and PfGDH aptamer modification on the surface of plastic optic fibers. After the addition of simulated malaria samples, the images of the light spots were captured by a smartphone camera and analyzed by ImageJ. software. Consequently, the smartphone-based aptasensor exhibited a LOD of 264 pM in spiked buffer and 352 pM in spiked serum. For the CK-MB, Zhang et al. [93] proposed a dual sensitization smartphone colorimetric strategy to detect CK-MB gathering rolling circle amplification (RCA) coils and Au tetrahedra. The platform demonstrated reasonable specificity with a low LOD of 0.8 pM and performed well in spiked human serum samples.

The above reports were all for the detection of a single target, Mahmoud et al. [94] proposed a novel system for the simultaneous determination of two proteins, interleukin-6 (IL-6) and thrombin, using corresponding antibodies and aptamers, respectively. The system was composed of a 3D-printed smartphone imager (Figure 4c) and lateral flow strips modified with QD labels thrombin aptamers [95] and interleukin-6 antibodies. Through RGB channels, colored pictures of lateral flow strips were split to achieve optical multiplexing, which reduced turnaround time. With the LOD of 100 pM for IL-6 and 3 nM for thrombin in spiked buffer solutions, the smartphone-based system was suitable for duplex detection, particularly in low-resource conditions.

**Figure 4 biosensors-12-00477-f004:**
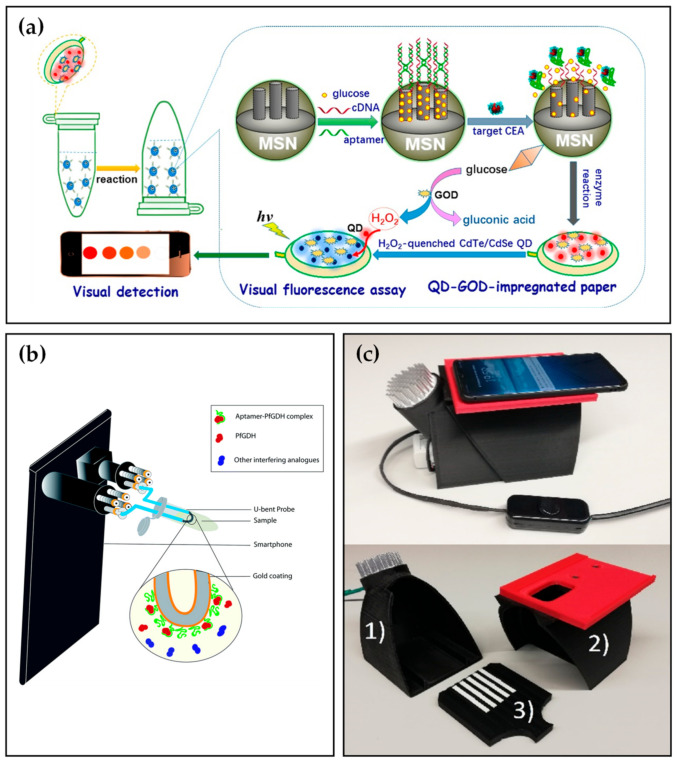
(**a**) Visual fluorescence detection mechanism of CEA biomarker based on paper-based analytical device (PAD). Reprinted with permission from Ref. [88], copyright 2017, American Chemical Society. (**b**) Schematic illustration of the smartphone-based optic fiber platform. Reprinted with permission from Ref [92], copyright 2020, RSC Publishing. (**c**) Smartphone dark box comprising (1) a bottom part fitted with a UV-LED light source, (2) a lid fitted with a replaceable smartphone adapter, and (3) a sample plate for seven lateral flow strips. Reprinted with permission from Ref [94], copyright 2021, Elsevier.

### 2.5. Bacteria Detection

Pathogenic bacteria cause various infectious diseases and kill millions of people each year [96]. Hence, to prevent and monitor diseases, it is crucial to detect the causative pathogens accurately and timely [97,98]. Specific recognition of Mycobacterium tuberculosis (M.tb) [99], Staphylococcus aureus (S. aureus) [100], and Salmonella Typhimurium (S. Typhimurium) [101] based on aptamers have been broadly reported. Recently, the detections of three above pathogenic bacteria based on smartphones and aptamers have been published for POCT detection. Li et al. [102] developed a reliable and cost-effective system based on enzyme-linked aptamers assay containing direct and indirect dot-blot assay (Figure 5A) for the detection of M.tb. Furthermore, an Android application performed colorimetric analysis on the photos captured by smartphone, and diagnostic reports were generated and transmitted through the Internet. This assay provided a lower limit of quantitation and a higher accuracy (Figure 5B) compared to traditional acid fast-staining assay, and greatly reduced the detection time compared to traditional bacterial culture.

For S. aureus, Shrivastava et al. [103] established a rapid, culture-free, smartphone-based fluorescence imaging platform for the quantitative detection of S. aureus based on the specific aptamer conjugated with fluorescent magnetic nanoparticles (FMNPs). As shown in Figure 5C, the S. aureus aptamer (Sap)-functionalized FMNPs were incubated with S. aureus, and then the fluorescence-tagged bacteria were spiked in complex, minimally processed samples. Next, the spiked samples were loaded into the bacterial detection cassette, followed by imaging using a smartphone fluorescence microscope comprised of a smartphone and a 3D-printed magnet-mounted holder. The top-right inset image represents the fluorescent intensity distribution of area-of-interest segmented from the captured image. By counting individual bacteria cells, the detection platform captured S. aureus cells efficiently from a peanut milk sample, with a minimum LOD of 10 cfu/mL within 10 min.

For S. Typhimurium, Man et al. [104] proposed a microfluidic colorimetric biosensor for on-site and rapid detection of Salmonella typhimurium based on aptamers and gold nanoparticles (AuNPs). Solutions mixing, reaction, and Salmonella detection occurred in the microfluidic chip (Figure 5D), where inlets (a, b, and d) were connected with a syringe pump to drive fluids, and a microvalve was used to control the flow direction of the fluids. Moreover, the colorimetric signals of the aggregated AuNPs were captured and processed by a smartphone imaging APP. The microfluidic aptasensor exhibited a low LOD of 6.0 × 10^1^ cfu/mL in spiked AuNPs solutions and may further improve the sensitivity and enable high-throughput detection with immunomagnetic bead technique and an optimized microfluidic device.

### 2.6. Multiple-Type Targets Detection

Multiple-type target simultaneous analysis detects two or more targets such as metal ions, small molecules, nucleic acids, proteins, glycoproteins, and bacteria using the same system. Gu et al. [105] developed a dual-signal sensing system of naked-eye colorimetry and magneto-electrochemical detection for the detection of ATP [106] and Hg^2+^ [107] based on aptamers. The electrochemical sensing unit included a homemade magneto-electrochemical detection module, a custom-built hand-held workstation, and a smartphone. The electrode chip-equipped electrochemical detection module is inserted into the embedded connector and controlled by a smartphone linked to the workstation via Bluetooth (Figure 6A). In the electrochemical unit, the LOD of ATP and Hg^2+^ were 5.203 pmol L^−1^ and 1.597 pmol L^−1^ (S/N = 3), respectively. More significantly, the dual-signal platform is well extensible, enabling on-demand rapid quantification of other targets by substituting specific aptamers.

Additionally, multiple-type target detection is often implemented based on lateral flow assays (LFA). For instance, Jin et al. [108] designed a lateral flow strip coupled with a smartphone-based portable device for the simultaneous and specific detection of multiple targets with three different aptamers. As depicted in Figure 6B, aptamers immobilized with three-colored upconversion nanoparticles (UCNPs) were used as capture probes and complementary DNA sequences as test and control probes in lateral flow aptamer assay. In the absence of targets (Salmonella, ochratoxin A, mercury ion (SE [109], OTA [110], and Hg^2+^ [111])), aptamers attached to UCNP probes are hybridized with the corresponding complementary DNA. In the presence of targets, the aptamers tend to bind the corresponding targets, thereby liberating UCNPs and decreasing fluorescence signals. Hence, the concentration of each target is inversely related to the corresponding color intensity of the colored band. Then, the detection results were read by the smartphone equipped with a portable device. Overall, the detection platform was further operated successfully in tap water samples within 30 min, offering a potential approach for convenient, simultaneous, sensitive detection in various fields.

Gong et al. [112] developed a UCNP-LFA platform that involved a similar homemade reader (Figure 6C(a,b)) of the above literature [108], realizing the quantitative of SE [109], OTA [113], and Hg^2+^ [111] by using custom-designed software. From the detection results in Figure 6C(c,d), the UCNP-LFA platform has a precision and accuracy comparable to the gold-standard method, and the correlation coefficients between them are over 0.992. Subsequently, the universality of this platform was successfully demonstrated by the detection of other targets involved in public health. The platform with quantitative and highly sensitive detection capability meets the requirement of clinical testing, which makes it highly promising for environmental pollution monitoring and food safety testing.

**Figure 6 biosensors-12-00477-f006:**
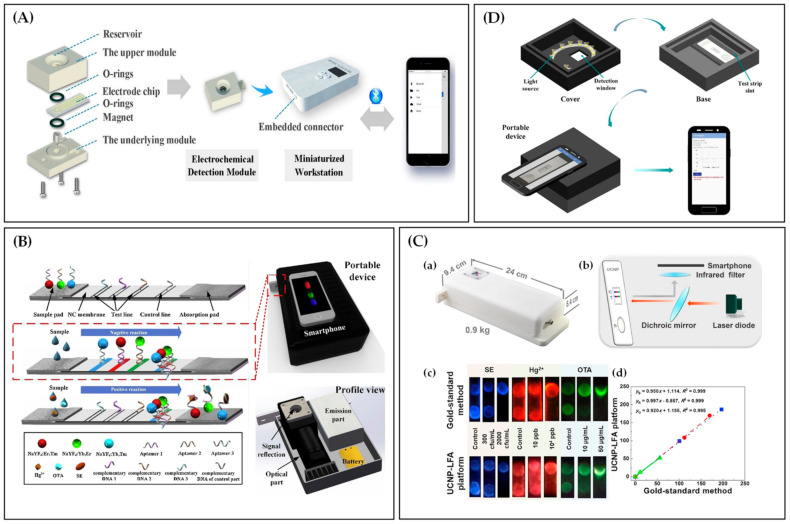
(**A**) Schematic of the electrochemical sensing unit. Reprinted with permission from Ref. [105], copyright 2020, Elsevier. (**B**) Schematic of lateral flow aptamer assay for simultaneous three kinds of targets detection. Reprinted with permission from Ref. [108], copyright 2018, Elsevier. (**C**) Photograph (**a**) and schematic (**b**) of UCNP-LFA reader. (**c**,**d**) Detection results of SE, Hg^2+^, and OTA using the UCNP-LFA platform and the gold-standard method. Reprinted with permission from Ref. [112], copyright 2019, Elsevier. (**D**) Detection result readout process based on a smartphone-based portable gadget and custom-designed app. Reprinted with permission from Ref. [114], copyright 2021, Elsevier.

Apart from DNA complementary sequences to the aptamers, antibodies are also acted as test and control probes in LFAs. Zhong et al. [114] constructed a multiple targets detection platform for rapid detection of Pb^2+^ [115], chloramphenicol [116], and β-lactoglobulin [117], based on aptamers and pregnancy test strips. In this LFA, aptamers were used as capture probes, while antibodies as the control probes. The pregnancy test strip shows two red lines when the target analytes are present and one red line when they are not. After the emergence of color, a portable gadget and a smartphone app were used to quantify the analytes (Figure 6D). Due to the maturity and ease of purchasing the pregnancy test strip, this detection platform based on LFA has a high potential for commercial application even in resource-limited settings.

## 3. Conclusions and Future Perspectives

Smartphone-based aptasensor is developing rapidly, which facilitates accurate, portable, rapid, and cost-effective detection. In this review, we discussed the detection methods of several substances using aptasensors on the platform of smartphones, including metal ions, small molecules, nucleic acids, proteins or glycoproteins, bacteria, and multiple-type targets. At present, most studies use aptamers as molecular probes of sensors to recognize and capture substances with their strong binding specificity. Smartphones, as data carriers, have been applied for colorimetric or electrochemical detection, in which data were sent to a smartphone via camera or Bluetooth for intelligent processing. Based on this method combined with smartphones, POCT can be used for medical, food, environment, and other fields, which makes the analysis and evaluation of detection results facility and convenient.

Aptasensors possess many advantages compared to other biosensors, such as enzyme biosensors and immunosensors. First, Aptamers can theoretically be selected in vitro for any particular target; thus, aptasensors have a more comprehensive range of applications in the detection field [118]. Second, high chemical stability and resistance to ambient temperature characterize the aptasensors, making them prosper in POCT. Third, the design of aptasensors is flexible due to the properties of aptamers with easy labeling and conformational changes upon target binding [119].

Smartphone-based aptasensor detection has broad prospects and great potential, but most aptasensor detection methods are still in the laboratory stage. Meanwhile, ELISA, electrochemiluminescence immunoassay (ECLIA), and other methods are primarily used in the clinical detection of antibodies. Aptamers have high specificity and trapping ability; however, we have not thoroughly investigated the properties of most aptamers [120]. In addition, the current research on aptamer is only to determine whether it can be used for substance detection, and the study of the modification and conformation change of aptamer is insufficient. The modification and assembly of aptasensors are also complicated, which takes time and effort to optimize the operating conditions. The cost of developing smartphone-based detection systems is very limited in remote areas with limited resources. This also requires professionals to develop, operate and expand the program, which gives users relative more difficulties in the process of participation [121].

Some designs need to be improved to develop smartphone-based aptasensors. For optical detection of smartphone-based aptasensor, further efforts are needed to reduce the volume and weight of the optical accessories by simplifying optical components and paths. Similarly, detectors of smartphone-based electrochemical aptasensor also need to be miniaturized. However, the fabrication of the integrated circuit chips is costly and complex. The built-in function modules of smartphones may replace them to eliminate hardware requirements [121].

In general, a smartphone-based aptasensor for diverse target detection has great advantages over the traditional method. Future efforts should focus on more advanced nanomaterials and molecular biotechnology to develop a more widely used and easily modified aptamer suitable for clinical application. In addition, it should also research photoelectric sensing systems and wireless communication equipment for smartphones to optimize the sensors. Furthermore, intelligent sensing technology will be combined with artificial intelligence, big data, the Internet of Things, cloud computing, and blockchain to become a more advantageous and promising detection technology. At last, costs should be reduced so that detection methods can be widely used in all regions to benefit all people. It is believed that smartphone-based aptasensors will play a more significant role in POCT in various fields after overcoming all challenges.

## Figures and Tables

**Figure 1 biosensors-12-00477-f001:**
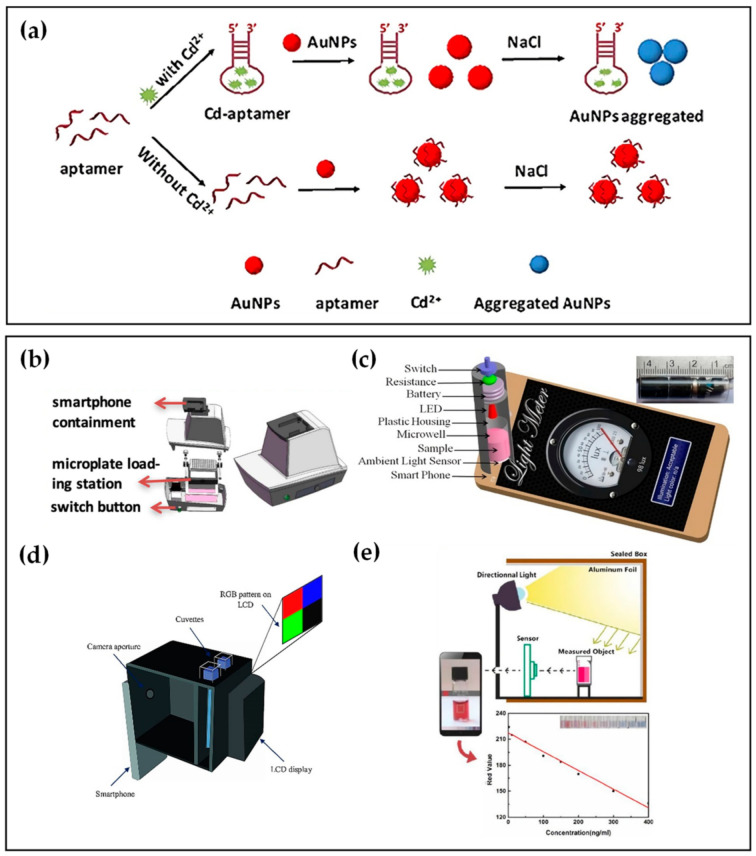
(**a**) Colorimetric detection principle based on AuNPs. Reprinted with permission from Ref. [48], copyright 2020, Elsevier. (**b**) Schematic illustration of the smartphone-based colorimetric system. Reprinted with permission from Ref. [48], copyright 2020, Elsevier. (**c**) Optical microwell reader picture and schematic of a microwell reader mounted on an Android smartphone. Reprinted with permission from Ref. [49], copyright 2016, MDPI, Basel, Switzerland, under the Creative Commons Attribution License. (**d**) Schematic of the designed smartphone-based colorimeter. Reprinted with permission from Ref. [50], copyright 2019, Elsevier. (**e**) Schematic illustration and the standard curve with different Cd^2+^ concentrations using the proposed smartphone-based colorimetric method. Reprinted with permission from Ref. [51], copyright 2019, MDPI, Basel, Switzerland, under the Creative Commons Attribution License.

**Figure 3 biosensors-12-00477-f003:**
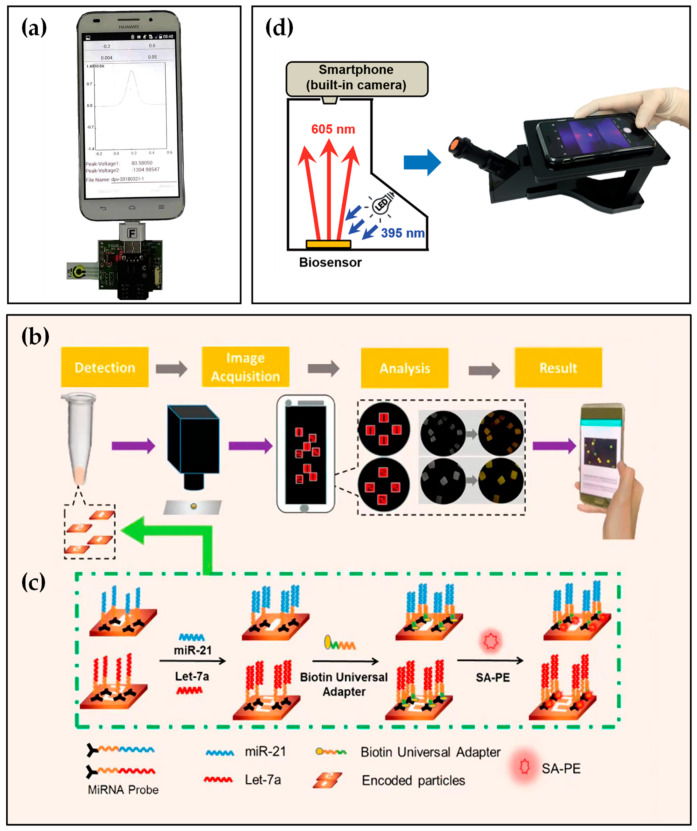
(**a**) Real image of the smartphone-based electrochemical biosensing system for detection of miR-21. Reprinted with permission from Ref. [20], copyright 2020, Elsevier. (**b**,**c**) Schematic illustration of the miR-21, let-7a simultaneous detection based on aptamer and smartphone. Reprinted with permission from Ref. [79], copyright 2019, American Chemical Society. (**d**) Internal illustration and appearance of self-made equipment with a smartphone. Reprinted with permission from Ref. [73], copyright 2018, Royal Society of Chemistry.

**Figure 5 biosensors-12-00477-f005:**
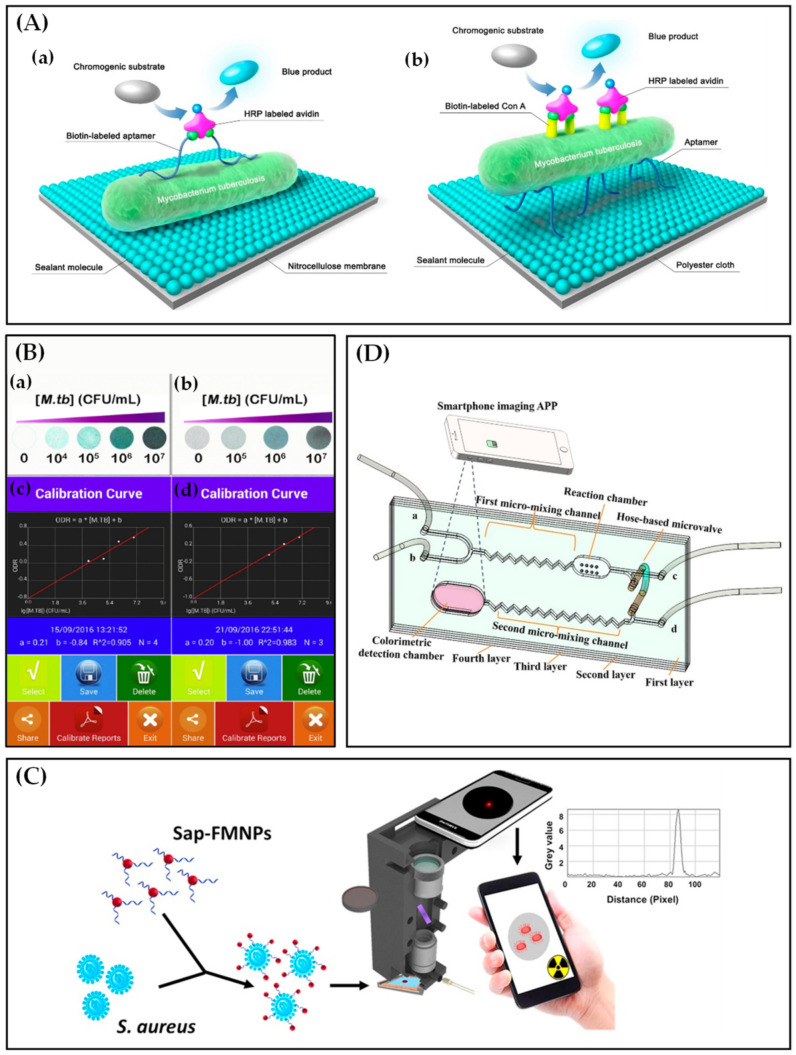
(**A**) Schematic illustrations of direct (**a**) and indirect (**b**) dot-blot assay. Reprinted with permission from Ref. [102], copyright 2018, Elsevier. (**B**) The color spots with varying M.tb concentration and application interfaces of the calibration curves in the direct (**a**,**c**) and indirect (**b**,**d**) dot-blot system, respectively. Reprinted with permission from Ref. [102], copyright 2018, Elsevier. (**C**) Schematic illustration of Staphylococcus aureus capture and quantitative detection in minimally processed samples. Reprinted with permission from Ref. [103], copyright 2018, Elsevier. (**D**) The schematic representation of the proposed microfluidic colorimetric biosensor based on a smartphone imaging APP for detecting Salmonella typhimurium. Reprinted with permission from Ref. [104], copyright 2021, Elsevier.

## Data Availability

Not applicable.

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
