# Peer review of "Applications of Smartphone-Based Aptasensor for Diverse Targets Detection"

_biosensors, 2022, doi:10.3390/bios12070477_

Round 1

Reviewer 1 Report

This manuscript reviews recent research advances in smartphone-based aptasensor for diverse targets detection, including metal ions, small molecules, nucleic acids, proteins or glycoproteins, bacteria, and multiple-type targets. Additionally, this manuscript introduces different strategies of smartphone-based aptasensors, and their values and outlook for applications in areas. Overall, this review is well organized and written, and smartphone-based aptasensor is attractive and promising for future applications. There are a little left for improvement before considering publication.

1.      Page 3 line 130 add a space before “ng/mL”.

2.      Page 5 table 1 add a space before “pg/mL”.

3.      Page 9 line 292 add a space before “nM”.

4.      Page 9 line 271 “ELISA” should be the full name with abbreviation.

5.      Page 14 line 418 the logic of “Although smartphone-based aptasensor detection has broad prospects and great potential, ELISA, electrochemiluminescence immunoassay (ECLIA) and other methods are mostly used in clinical detection of antibodies, while most of the aptasensor detection methods are still in the laboratory stage.” is not very clear and should be reviewed.

6.      In the outlook section, the description is a little general, can author provide additional points of view about the applications of specific aspects?

Author Response

Dear reviewer,

Thanks very much for taking the time to review this manuscript. I really appreciate all your comments and suggestions. Please find itemized responses, revisions, and corrections in the re-submitted files below.

  1. Page 3 line 130 add a space before “ng/mL”.
  2. Page 5 table 1 add a space before “pg/mL”.
  3. Page 9 line 292 add a space before “nM”.
  4. Page 9 line 271 “ELISA” should be the full name with abbreviation.

Response to comments 1-4:  All of these errors have been corrected; we are thankful to the referee for careful reading. We also revised other typos and highlighted them in yellow as well.

  1. Page 14 line 418 the logic of “Although smartphone-based aptasensor detection has broad prospects and great potential, ELISA, electrochemiluminescence immunoassay (ECLIA) and other methods are mostly used in clinical detection of antibodies, while most of the aptasensor detection methods are still in the laboratory stage.” is not very clear and should be reviewed.

We have changed the sentence

“Although smartphone-based aptasensor detection has broad prospects and great potential, ELISA, electrochemiluminescence immunoassay (ECLIA) and other methods are mostly used in clinical detection of antibodies, while most of the aptasensor detection methods are still in the laboratory stage.”

to now read

“Smartphone-based aptasensor detection has broad prospects and great potential, but most aptasensor detection methods are still in the laboratory stage. Meanwhile, ELISA, electro-chemiluminescence immunoassay (ECLIA), and other methods are primarily used in the clinical detection of antibodies.”

  1. In the outlook section, the description is a little general, can author provide additional points of view about the applications of specific aspects?

On Page 15 Line 432, we now briefly discuss electrolyte methodology studies on translation rate by adding

“Some designs need to be improved to develop smartphone-based aptasensors. For op-tical detection of smartphone-based aptasensor, further efforts are needed to reduce the volume and weight of the optical accessories by simplifying optical components and paths. Similarly, detectors of smartphone-based electrochemical aptasensor also need to be miniaturized. However, the fabrication of the integrated circuit chips is costly and complex.  The built-in function modules of smartphones may replace them to eliminate hardware requirements [123].”

123. Zhang, D.; Liu, Q. Biosensors and bioelectronics on smartphone for portable biochemical detection. Biosens Bioelectron 2016, 75, 273-284, doi:10.1016/j.bios.2015.08.037.

Reviewer 2 Report

Aptasensors are broadly developed to detect various substance. Most aptasensors are developed based on precise instruments which are inconvenient and uneconomic. Some studies were turned to combining portable device such as smartphone with biosensors for data gathering. In this review, the authors summarized current applications of smartphone-based aptasensors, which may provide a direction for other researchers on the design of portable aptasensors. The illustrations in this review are well-organized, but the narration in the text is plain. Besides, following concerns should be noted.

In the Abstract, the authors claimed “Aptamers are nucleic acid-based target recognition elements for biomolecules detection” that is not correct. First, aptamer includes both nucleic acid aptamer and peptide aptamer; second, aptamers are not expressly applied for biomolecules detection, they are also used in non-biomolecule detection.

Line 38-39, Therefore, POCT with low cost and easy operation are urgently needed to be developed.

Line 55-56, not all apamers are selected by SELEX, some other technologies can be used for aptamer selection, please revise it.

Line 58-59, please specify the new SELEX strategies

Line 70, detect_five?

Line 81-83, please specify the novel SELEX strategy

Line 122, please uniform the units (1nM-?nM)

Line 124, by Xu et al.

Line 133, Figure 1a, to make the figure visual, could please authors change the globule of AuNPs to purple or grey when they are aggregated, while red when they are dispersed. Please move down the legend to separate the legend and you flow chart, otherwise, it is confused. The same issue is advised in Figure 2a.

Line 204, the authors claimed that DNAzymes are single-stranded DNA aptamers that display a catalytic activity. DNAzymes are single-stranded DNA, but cannot be called aptamers, they have biological functions besides target-binding (please refer the statement in Nucleic Acids Research, Volume 48, Issue 7, Pages 3400–3422)

In the main text, the authors jointed multiple research reports together but there is less discussion regarding to their pros and cons. For example, line 211-230, smartphones are used in those reports, but those may employed other biosensor technologies otherwise aptasensors, could please authors discuss the feasibility of aptasensor in that field?

In the Conclusions and future perspective, the authors’ view on the future development of smartphone-based aptasensor is lacking. With the abundant knowledge of authors in this field, could please authors point the current shortage and specify how to develop the smartphone-based aptasensor detection technology. Readers and researchers are expected to see more clear direction in the further development of this technology by reading this review. The final aim of the researchers is to translate this technology into application products.

Reviewer 3 Report

This is an interesting review where the authors present a good summary of advancements in smartphone-based aptasensors, characterized by different detection  methods of several substances relying on aptamers as molecular probes of  sensors to recognize and capture variuos target analytes with their strong binding specificity. Review is well-written and very clear.

I have no major concerns, I just suggest to specify some acronyms, such as PDDA (page 3, line 123) and NSET (page 5, line 173) and to swap panel e and panel g in figure 2 (there was a misconstruction!).

Overall, I recommend publication of this review.

Author Response

Dear reviewer,

Thanks very much for taking your time to review this manuscript. I really appreciate all your comments and suggestions. All of these errors have been corrected. Please find my revisions and corrections in the re-submitted files.

PDDA (page 3, line 123) and NSET (page 5, line 173) and to swap panel e and panel g in figure 2 (there was a misconstruction!).

All errors have been corrected and we are thankful to the referee for careful reading.